# Multimodal Prehabilitation for Patients with Crohn’s Disease Scheduled for Major Surgery: A Narrative Review

**DOI:** 10.3390/nu16111783

**Published:** 2024-06-06

**Authors:** Camilla Fiorindi, Francesco Giudici, Giuseppe Dario Testa, Lorenzo Foti, Sara Romanazzo, Cristina Tognozzi, Giovanni Mansueto, Stefano Scaringi, Francesca Cuffaro, Anita Nannoni, Mattias Soop, Gabriele Baldini

**Affiliations:** 1Department of Health Science, University of Firenze, Azienda Ospedaliero Universitaria Careggi, Largo Brambilla 3, 50139 Florence, Italy; sara.romanazzo@unifi.it (S.R.); tognozzic@aou-careggi.toscana.it (C.T.); giovanni.mansueto@unifi.it (G.M.); francesca.cuffaro@unifi.it (F.C.); nannonia@aou-careggi.toscana.it (A.N.); gabriele.baldini@unifi.it (G.B.); 2Multimodal Prehabilitation Center, Azienda Ospedaliera Universitaria Careggi, Largo Brambilla 6, 50135 Florence, Italy; giuseppedario.testa@unifi.it (G.D.T.); lorenzosanto.foti@unifi.it (L.F.); 3Department of Experimental and Clinical Medicine, University of Florence, Largo Brambilla 6, 50135 Florence, Italy; francesco.giudici@unifi.it (F.G.); scaringis@aou-careggi.toscana.it (S.S.); 4Division of Geriatric and Intensive Care Medicine, University of Florence, Azienda Ospedaliero Universitaria Careggi, Largo Brambilla 3, 50139 Florence, Italy; 5Section of Anesthesiology and Intensive Care, University of Florence, Largo Brambilla 3, 50139 Florence, Italy; 6Department for IBD and Intestinal Failure Surgery, Karolinska University Hospital, SE 177 76 Stockholm, Sweden; mattias.soop@ki.se

**Keywords:** Crohn’s disease, Enhanced Recovery After Surgery (ERAS), prehabilitation, multimodal, nutrition, surgery

## Abstract

Approximately 15–50% of patients with Crohn’s disease (CD) will require surgery within ten years following the diagnosis. The management of modifiable risk factors before surgery is essential to reduce postoperative complications and to promote a better postoperative recovery. Preoperative malnutrition reduced functional capacity, sarcopenia, immunosuppressive medications, anemia, and psychological distress are frequently present in CD patients. Multimodal prehabilitation consists of nutritional, functional, medical, and psychological interventions implemented before surgery, aiming at optimizing preoperative status and improve postoperative recovery. Currently, studies evaluating the effect of multimodal prehabilitation on postoperative outcomes specifically in CD are lacking. Some studies have investigated the effect of a single prehabilitation intervention, of which nutritional optimization is the most investigated. The aim of this narrative review is to present the physiologic rationale supporting multimodal surgical prehabilitation in CD patients waiting for surgery, and to describe its main components to facilitate their adoption in the preoperative standard of care.

## 1. Introduction

The risk of undergoing surgery in patients with Crohn’s disease (CD) is approximately 15–50% within 10 years of the diagnosis [1].

Preoperative nutritional status correlates with postoperative outcomes [2]. The aetiology of malnutrition in patients afflicted with Inflammatory Bowel Disease (IBD) encompasses multifaceted elements, including reduced dietary intake, intestinal malabsorption and protein loss, and elevated energy requisites due to systemic inflammation. However, a precise prevalence of malnutrition in this population remains unclear [3]. In particular, published rates of malnutrition in CD ranges widely, from 20% to 85% [4,5]. Therefore, it is recommended that nutritional status should be evaluated and optimized before elective surgery, whenever possible [6], and that nutritional care is ensured throughout the entire perioperative period [7].

It has been shown that preoperative nutritional status is an independent risk factor for anastomotic leakage and that preoperative nutritional therapy decreases the risk of anastomotic leakage in CD [8]. These results were also confirmed by a Cochrane review of preoperative nutritional support in patients awaiting gastrointestinal surgery for inflammatory bowel diseases and other gastrointestinal pathologies [7,9]. Furthermore, results from other meta-analyses have reported that preoperative nutrition supplementation reduces postoperative complications in patients with CD by 74% (Odds ratio = 0.26, 95% confidence interval (CI): 0.07–0.99, *p* < 0.001) [10]. Based on these data, the European Society of Clinical Nutrition and Metabolism (ESPEN) guidelines have recommended assessment and optimization of nutritional status, including nutritional prehabilitation, in the context of an Enhanced Recovery After Surgery (ERAS) setting (grade of recommendations: extrapolated evidence from studies, 96% strong consensus) [11].

The term “prehabilitation” characterizes a broad range of interventions that primarily seek to improve the preoperative medical, physical, nutritional, and psychological status. In some publications, the term is used to refer exclusively to interventions related to physical exercise. Here, the broader scope of the term will be discussed.

In this clinical context, nutritional interventions aim at correcting malnutrition, optimizing body composition, ensuring adequate caloric and protein intake, controlling blood glucose, providing immunonutrients, managing gastrointestinal symptoms, and overcoming barriers preventing food intake. Preliminary evidence suggests that nutritional prehabilitation is effective in increasing preoperative body weight, body mass index, and fat free mass index in both CD and ulcerative colitis patients, thus attenuating the loss of lean body mass observed postoperatively [12]. A minimum period of 7–14 days of nutritional intervention is recommended by ESPEN to positively impact postoperative outcomes. However, based on individual needs, nutritional optimization might be extended up to 6–8 weeks to better prepare the surgical patient to withstand the surgical stress and further improve postoperative outcomes [11,13]. In fact, a prolonged nutritional prehabilitation (approximately 3 months) seems to optimize the body composition of patients waiting for elective surgery and could be a useful strategy to mitigate the catabolic effect of the surgical stress response and the resulting loss of lean body mass, especially if implemented in an ERAS setting [12]. In fact, this integrated approach has been shown to facilitate the recovery of bowel function and shorten hospital stay [12]. A recent systematic review has demonstrated that a trimodal prehabilitation program, including exercise and inspiratory muscle training and nutritional and psychological support improves the preoperative functional capacity and reduces postoperative complication rates after major abdominal surgery [14]. Furthermore recent studies showed an association between physical activity, resilience, and quality of life in IBD patients [15,16]. Nevertheless, studies evaluating the efficacy of trimodal prehabilitation in CD surgical patients are currently lacking.

The aim of this narrative review is to present the physiologic rationale supporting multimodal surgical prehabilitation in CD patients waiting for surgery and describe its main components.

## 2. Nutritional Optimization

### 2.1. Nutritional Assessment and Monitoring

In CD patients, malnutrition is the result of inadequate nutritional intake, malabsorption, and increased energy expenditure due to active inflammation [11,17].

Assessing and enhancing nutritional status is a key element of surgical prehabilitation. First, nutritional screening should be performed in all patients undergoing surgery to identify patients at risk of malnutrition and those who are malnourished [11,17]. Several validated screening tools can be used to detect the risk of malnutrition in the CD population, such as the Malnutrition Universal Screening Tool (MUST), the Nutrition Risk Screening (NRS-2002), the Malnutrition Inflammation Risk Tool (MIRT), and the Saskatchewan IBD Nutrition Risk Tool (SaskIBD-NR) (Table 1). An optimal validated screening tool specific for IBD patients has not yet been identified [18]. Recently the Inflammatory Bowel Disease-Nutritional Screening tool (NS-IBD) was developed to screen IBD patients [19].

Several parameters, such as the patient clinical and social history, anthropometric measures, dietary intake, and nutritional and inflammatory biomarkers, should be evaluated during the initial nutritional assessment. This information helps to identify high-risk patients, as unintentional weight loss (UWL), body mass index (BMI), and reduction in dietary intake are strong predictors of nutritional risk [17].

An unintentional weight loss of more than 10% from baseline within the 6-month period preceding surgery has been reported in 23–54% of CD patients scheduled for intestinal resection [1]. Numerous investigations have revealed an evident association between recent UWL and surgical complications in CD patients [20,21,22,23,24]. According to a systematic review and meta-analysis, patients with CD were shown to have inadequate energy intake, especially during active disease. Furthermore, protein consumption was considerably lower than those of healthy controls, thus exposing patients to a greater nutritional risk [25]. A recent study has evaluated the dietary intake of IBD patients scheduled for surgery and compared it with the dietary reference values (DRV) derived from the Italian population (LARN), and with those indicated by the ESPEN guidelines for clinical nutrition in IBD. It was noted that IBD patients had inadequate intake of proteins, *n*-3 PUFA, fibre, iron, calcium, potassium, magnesium, zinc, vitamin D, and vitamin B12 with respect to the DRV. Oral intake was not influenced by gender, IBD subtype, duration of disease, and previous surgery. Moreover, it was found that in CD patients fistulizing behaviour negatively influenced oral intake [26].

BMI is a simple and fast tool used to assess nutritional status. Low preoperative BMI was associated with a higher risk of postoperative infectious complications and anastomotic leakage, reoperations, longer hospital stays, and higher mortality [27,28]. On the other hand, patients with an increased BMI are also at risk of surgical complications, especially infectious morbidity. Furthermore, obese patients are at a higher risk of complications than those who are overweight [29]. However, BMI has many limitations, as it does not provide any information about patient body composition. In fact, patients with identical BMI might have completely different lean body mass [30], and patients with different BMI might have similar lean body mass. Therefore, body composition assessment tools should be always used during nutritional assessment to identify patients with limited skeletal mass, despite normal BMI or total body weight. Notably, altered body composition is a key predictor of poor outcomes and surgical complications for CD patients [17,31].

If intestinal malabsorption is suspected, serum parameters as folic acid, vitamin B12, vitamin D, iron, zinc, magnesium, and selenium should be evaluated [32]. Patients with previous ileocecal resection with a vegan regimen or avoiding meat and dairy are at higher risk of vitamin B12 deficiency [11]. The prevalence of B12 deficiency in CD ranges from 5.6 to 38% [33]. Overall, anaemia is prevalent in up to 78% of CD patients requiring a surgical intervention and is associated with postoperative complications; furthermore, its severity is associated with poor postoperative outcomes [34,35]. The interpretation of iron serum level could be influenced by inflammatory status. In fact, in patients with IBD, anaemia of chronic disease often co-exists with iron deficiency anaemia. A case-control study showed significantly lower iron levels in CD patients compared to healthy patients, independently from their disease status (active or in remission) [36,37].

The ESPEN guidelines on clinical nutrition in IBD suggest monitoring serum calcium and 25(OH) vitamin D in patients with active disease and in those who are treated with steroids to help prevent low bone mineral density. Moreover, in geriatric CD patients, due the positive association between low vitamin D and loss of muscle mass in individuals > 65 years old [11], measuring 25(OH) vitamin D can identify patients who might benefit from Vit D supplementation to further optimize their muscle mass [38].

In IBD, the assessment and the interpretation of some serum parameters could be affected by inflammation. In fact, biochemical nutrients such as albumin may not be indicative of malnutrition in case of active CD. Serum albumin levels are lower in CD patients than in the healthy population and are inversely linked with disease activity, likely as a result of the chronic inflammatory condition [39]. Low serum albumin concentrations are associated with an increased risk of postoperative complications in CD patients [40,41], likely as a result of more severe systemic inflammation rather than an impaired nutritional state.

Although several validated tools exist to assess nutritional state in those patients identified by screening to be at risk of malnutrition, a gold standard method is still lacking. Recently, the Global Leadership Initiative on Malnutrition (GLIM) criteria were developed to address this, with the intent to present a uniform malnutrition diagnosis in clinical practice. According to GLIM criteria, the diagnosis of malnutrition is based on the presence of at least one phenotypic criterion (UWL; low BMI; reduced free fat mass (FFM)) and at least one etiologic criterion (reduced food intake or absorption; inflammation). A recent study showed that 34% of CD patients were identified as malnourished using the GLIM criteria [42]. According to the etiologic criterion, malnutrition in IBD seems linked to inflammation and malabsorption, even in absence of reduced food intake.

Sarcopenia, which is characterized by low lean muscle mass along with either poor muscle strength or low physical performance, is a common condition among CD patients. Sarcopenia can occur in both underweight and overweight IBD patients [43]. A recent systematic review of 18 studies found that the prevalence of sarcopenia among CD patients, ranges from 16 to 100%, and is higher than 50% in more than half of the included studies. This large variability might be explained by the lack of a universal definition of sarcopenia, by the anthropometric and socio-cultural heterogeneity of the population of interest, alongside with the high variability of muscle mass indices and thresholds used to identify sarcopenic patients [44]. The European Working Group on Sarcopenia in Older People (EWGSOP) identifies low muscle strength, and low muscle quantity and quality as mandatory features to diagnose sarcopenia; physical performance status is used to quantify the severity of sarcopenia [45]. Among several measures to assess body composition in IBD patients, Skeletal Muscle Mass Index (SMI) seems to be the most frequently used parameter to quantify muscle mass [31]. IBD patients might require urgent surgery, and in this clinical context it is strongly associated with postoperative complications [46]. In patients with IBD, sarcopenia has been identified as an independent predictor for both surgical complications and/or adverse outcomes, such as poor quality of life and inadequate response to medical therapy [47]. Since sarcopenia can be reversed, nutritional support and education provided by trained dietitians, together with exercise, are essential to reduce perioperative risks and optimize surgical outcomes [47].

### 2.2. Nutritional Intervention

Preoperative nutritional intervention in CD patients has multiple purposes: improving nutritional status, optimizing body composition, ensuring adequate caloric and protein intake, controlling blood glucose, providing immunonutrients, managing gastrointestinal symptoms, overcoming barriers preventing food intake, withstanding the catabolic impact of surgery, and supporting the energetic cost of exercise interventions in the context of a multimodal prehabilitation program. The ultimate goal is to reduce postoperative complications and facilitate surgical recovery [13,17] (Figure 1).

#### 2.2.1. Enteral Nutrition (EN)

Whenever possible, nutritional support should be provided primarily through the enteral route administered orally or by a feeding tube [13,48]. It is recommended when patients have an inadequate oral intake or if they are unable to ensure intake levels above 50% of their recommended daily needs for more than 7 days [13].

In CD patients, preoperative EN reduces postoperative complications by 74% compared to CD patients receiving standard of care without nutritional support [10]. Two recent retrospective studies also confirmed these finding [2,49]. Abdalla et al. suggested that intra-abdominal septic complications (IASCs) and the need for defuncting stomas might be reduced with preoperative enteral nutrition [50].

Preoperative exclusive enteral nutrition (EEN) is a therapeutic regimen in CD based on the exclusive use of a nutritionally complete liquid formula as the only source of nutrition. Through mechanisms that are not yet fully explored, EEN reduces inflammation and disease activity in CD, and is recommended as the first-hand therapy in paediatric CD [11]. In the preoperative setting, it potentially enhances surgical outcomes in adults with CD [50]. A significant decrease in overall postoperative complications [51,52,53], systemic inflammation, postoperative abscesses, and anastomotic leakage [53,54] was also observed in cohort studies that examined the impact of preoperative EEN in CD patients [55,56]. In one study, preoperative EEN was effective in reducing the risk of postoperative intra-abdominal septic complications (IASCs) [54,57]. These results were subsequently confirmed by Zhu et al. [55]. One prospective trial evaluated the impact of preoperative EEN with a polymeric diet enriched with transforming growth factor-beta 2 on postoperative outcomes in both low-risk and in high-risk CD patients (defined by the presence of obstructive symptoms, steroid treatment, preoperative weight loss > 10%, and perforating CD) who underwent gastrointestinal surgery. It was found that the incidence of postoperative complication was comparable between the two groups [23]. According to one systematic review, the preoperative role of EEN in CD patients is still unclear, as good quality studies and large prospective trials are lacking [58]. However, if these preliminary and limited data are confirmed, preoperative EEN could be a beneficial nutritional strategy to reduce the risk of postoperative complications in high-risk malnourished CD patients, including early recurrence, within 6 months from surgery [53,56]. The European Crohn’s and Colitis Organization (ECCO) practice position states that pre-operative EEN in patients with stricturing or penetrating CD improves nutritional status and may reduce postoperative complications [23]. Multidisciplinary discussion about the optimal treatment duration and route of administration is advised.

A low residue diet with modified consistency or EEN via a feeding tube ending distal to the obstruction (post-stenosis), when possible, can be advised for CD patients who also have intestinal strictures or stenosis with obstructive symptoms [11].

#### 2.2.2. Parenteral Nutrition (PN)

PN may be used to meet the recommended daily energy and protein requirements only when EN is contraindicated, ineffective, or poorly tolerated, either in combination with EN (partial parenteral nutrition PPN) or as a total parenteral nutrition (TPN). EN is generally preferred over PN due to its preservation of gut function, reduced risk of infections, cost-effectiveness, and improved long-term tolerance [59]. TPN is usually recommended in CD patients with prolonged intestinal failure [11,13]. The administration of preoperative PN frequently occurs in patients with perforating CD complicated by malnutrition [50].

There is little research on how PN affects postoperative complications in CD. One recent retrospective study compared the effect of TPN on postoperative complications between CD patients who received and did not receive TPN. Despite patients treated with TPN being more severely ill with inadequate oral intake, complications rates were comparable between the two groups [60]. A significant reduction in anastomotic leakage, postoperative complications, and non-infectious complications in CD patients receiving perioperative TPN has also been found in some studies [22,61].

#### 2.2.3. Energy and Nutritional Requirements

ESPEN practical guidelines [8] recommends that energy delivery should be 30–35 kcal/kg/day since the energy requirements of patients with IBD are similar to those of the healthy population. If different energy needs are suspected, individual energy requirements should be determined using indirect calorimetry, especially in the context of a multimodal prehabilitation program, during which the metabolic cost of exercise should be always estimated and covered.

ESPEN practical guidelines [8] recommend a protein intake of 1.2–1.5 g/kg/d in adults with active IBD, higher than those recommended for patients without IBD. This recommendation is based on the chronically poor or unbalanced dietary intake of IBD patients, increased rates of protein turnover, gut loss of nutrients during phases of active disease with consequent malabsorption, and on the effect of disease treatments, such as corticosteroids.

In the context of a multimodal prehabilitation program, supplementing patients with whey protein might potentiate the anabolic effect resistance exercise and increase preoperative muscle mass [62]. Although this type of supplementation has not yet been studied in CD patients, oral supplementation with 20 g of whey protein has been shown to be clinically effective in improve preoperative functional walking capacity in patients with colorectal cancer [63].

In case of iron deficiency anaemia and/or anaemia of inflammatory disease, intravenous (IV) iron is likely to correct anaemia more quickly than oral supplementation as recommended by the ECCO guidelines [6].

Vitamin D combined with whey protein improves muscle function and mass in older adults, with or without exercise [64,65]. In case of deficit, vitamin D supplementation of 600–800 international units (or even higher doses) might prevent or treat low muscle mass of adult patients [66].

## 3. Functional Capacity and Muscle Strength Optimization

### 3.1. Assessment and Monitoring

Functional capacity refers to an individual’s ability to respond effectively to physiological stress, and in the surgical setting, to the prolonged increased of aerobic metabolism (oxygen consumption, VO_2_) observed after major surgery [67]. There are several similarities between the physiological stress, induced for example by exercise, and the stress experienced during the perioperative phase [68]. Indeed, surgical patients have an increased heart rate, cardiac workload, and oxygen utilization at the cellular level to maintain adequate tissue oxygenation and ensure organ function and healing. Postoperatively, these changes require an overall increase in oxygen consumption, and surgical patients who are unable to attain these requests have higher morbidity, mortality rates [69,70], and impaired functional recovery [71]. For example, inadequate tissue oxygenation is a critical condition that significantly affects the occurrence of anastomotic leaks after colorectal surgery [72]. Reduced preoperative functional capacity is a well-established risk factor for postoperative complications in patients undergoing colorectal surgery. IBD appears to be associated with a reduced preoperative functional capacity as compared with other colorectal diseases [73,74,75,76]. Furthermore, preliminary data have shown that patients with IBD have impaired recovery of heart rate after physiological stress, and this may further increase perioperative risks [77]. These data support the physiologic rational to employ preoperative multimodal corrective strategies, such as prehabilitation, aiming at optimizing preoperative functional [78]. However, there are currently few but encouraging data on the role of prehabilitation in patients with IBD [79]. The recommendations of the American Heart Association/American College of Cardiology (AHA/ACC) and the European Society of Cardiology/European Society of Anaesthesia (ESC/ESA) emphasize the importance of measuring functional capacity before non-cardiac surgery. In the real world, the assessment of functional capacity usually involves a subjective estimate by the physician, based, for example, on the subject’s reported ability to climb two flights of stairs (equivalent to four metabolic equivalents). Previous research has shown that oxygen utilization while climbing two flights of stairs is equivalent to four metabolic equivalents, corresponding to the oxygen consumption threshold necessary to minimize the occurrence of postoperative complications minimized [80]. However, the Measurement of Exercise Tolerance before Surgery (METS) study, a multicentre prospective cohort study, revealed that unstructured subjective assessment of functional capacity is not sensitive for identifying patients with poor preoperative cardiorespiratory and metabolic reserve (sensitivity 19.2%) and fails to accurately identify patients at high risk of postoperative morbidity [81]. Although subsequent trials have shown that unstructured subjective assessment of functional capacity might predict major cardiovascular complications and mortality, especially in a surgical population with a high cardiovascular risk, more objective methods such as the cardiopulmonary exercise test (CPET), the Six-Minute Walking Test (6MWT), and the Duke Activity Status Index (DASI) have shown better accuracy for quantifying functional capacity and predict postoperative adverse outcomes [82] (Table 2).

The CPET is widely recognized as the gold standard for measuring functional capacity. Through the measurement of peak/maximum oxygen consumption (VO_2-peak_, VO_2-max_) and anaerobic threshold (VO_2-AT_), it allows us to investigate the relationship between the performance of the cardiopulmonary system and the way cells utilize oxygen [81]. Previous studies have shown an association between decreased preoperative VO_2-peak_ and VO_2-AT_ and the risk of postoperative complications [96,97]. Although sparse evidence suggest that patients with VO_2-peak_ less than 15 mL/kg/min or an VO_2 AT_ less than 10–12 mL/kg/min are at high-risk of developing overall postoperative complications [81], values above these thresholds are more useful to rule out the risk of developing complications as well as in-hospital and 30-day mortality after non-cardiopulmonary surgery [98,99]. In the context of a multimodal prehabilitation program, the CPET is not only useful to stratify preoperative risk, but also to tailor the exercise prescription and to monitor the effectiveness of the prehabilitation intervention. Studies on cardiac rehabilitation have shown that CPET plays a crucial role in adapting the intensity and volume of exercises, for example through the assessment of ventilatory thresholds [85]. Similarly, CPET can be useful to verify the efficacy of exercise-prehabilitation by measuring preoperative changes of VO_2-peak_ and/or VO_2 AT_ [100,101]. However, as CPET requires time and dedicated resources, other measures, such as the 6MWT or the DASI, which have respectively demonstrated good and moderate predictive ability to assess functional capacity, are often preferred [102,103]. The 6MWT is a submaximal exercise test that requires participants to walk at a brisk pace continuously for a period of 6 min on a straight and level pathway 25 m in length [104]. The main objective of the test is to quantify functional capacity by measuring the distance travelled during the walking exercise. Vital parameters (oxygen saturation, heart rate, respiratory rate, and blood pressure) and perception of exertion (e.g., Borg scale) are measured before and after the test to determine the physiological and subjective response to exertion. Like the CPET, the 6MWT can provide meaningful information. First, the 6MWT has a prognostic value, as it is known that a preoperative 6-min walking distances < than 400 metres predicts postoperative complications, their severity, and impaired postoperative functional recovery after colorectal surgery [91,105]. Second, the 6MWT may be useful for exercise prescription, as it allows recording the chronotropic response to exercise and the perception of exertion using subjective methods such as the Borg scale [106,107]. Third, the 6MWT can be useful for monitoring the effectiveness of the prehabilitation treatment. It was shown, for example, that a 19-m improvement at 6MWT after a 4-week rehabilitation program was predictive of a reduced risk of postoperative complications in colorectal surgery [105]. In a sub-analysis of the METS study, the 6MWT was shown to have comparable or higher predictive power than the CPET in predicting post 1-year disability free survival, 30-day death or myocardial infarction, and 1-year mortality.

The DASI is a 12-component questionnaire with a range of values between 0 and 58.2 that has been validated to assess the physical activity level of adults [108]. Recently, the DASI has demonstrated its usefulness for the prognostic stratification of candidates for major surgery, especially for predicting cardiovascular complications. Indeed, a secondary analysis of the METS study showed that a DASI score of less than 34 identifies patients at increased risk of myocardial damage, myocardial infarction, serious complications, and the development of new disabilities [109]. In other studies, the prognostic power of the DASI for identifying patients at risk of adverse postoperative outcomes was superior to those of the 6MWT and of the CPET [110]. Although the DASI score could be used to calculate the predicted VO_2-peak_ (predicted VO_2-peak_ = 0.43 × DASI + 9.6) [108], the METS trial showed that a DASI score of 34 overestimates functional capacity by 2 METs, thus categorizing patients with poor functional capacity as “fit” for surgery. Moreover the formula used to estimate the VO_2-peak_ was validated in a general population that might be functionally different from the surgical population [108]. Argillander et al. proposed the integration of different physical tests assessing both aerobic fitness and muscular strength to provide a more in-depth assessment of the patient’s physical status, and not relying solely on a single preoperative test [111]. Indeed, patients’ ability to perform the basic activities of daily living and overcome the surgical stress does not depend exclusively on functional capacity [112,113]. Indeed, various factors, including age, comorbidities, cognitive status, skeletal muscle mass and quality (sarcopenia), nutritional status, and frailty, have been shown to influence directedly or indirectly functional capacity and thus negatively impact on postoperative outcomes. Moreover, malnutrition, frailty and sarcopenia frequently coexist in the surgical population [114]. Given the complexity in evaluating preoperative functional status, and considering the increasing ageing of the IBD population, supplementing routine preoperative assessment with the evaluation of sarcopenia, malnutrition, and frailty might provide additional valuable information to better prepare IBD patients to major surgery. In fact, it is widely recognized that sarcopenia, malnutrition, and frailty have are significant risk factors for unfavourable postoperative outcomes after abdominal surgery, [115,116,117]. The gait speed, hand strength test, Timed Up and Go (TUG), 5-time chair standing test, Short Physical Performance Battery (SPPB), and Fried Phenotype Criteria are validated tools for identifying and assessing physical frailty, and most of them are also included in the current algorithm for diagnosing and establishing the severity of sarcopenia [45,118,119,120,121]. Finally, recent evidence supports the notion that frailty and sarcopenia play a crucial role as predictors of mortality, therapeutic failure, and surgical complications specifically in patients with IBD [122,123].

### 3.2. Exercise Intervention

Common physical interventions used in the context of a prehabilitation program are aerobic and resistance exercise training. In CD patients at high-risk of pulmonary complications, inspiratory muscle training maybe also beneficial.

As any medication prescribed requires a precise and individualized posology, exercise prescription should follow the same principles. The American College of Sports Medicine recommends using the FITT-VP (Frequency, Intensity, Timing, Type plus Volume and Progression) principles when prescribe exercise training [124,125].

#### 3.2.1. Aerobic Exercise Training

Aerobic training, stimulating the cardiovascular system, is the most effective intervention to improve and optimize the relationship between Oxygen Delivery (DO_2_) and Consumption (VO_2_). There are two main types of aerobic training: High-Intensity Interval Training (HIIT), alternating short periods of high and low intensity workout, and Moderate Intensity Continuous Training (MICT), which consists of exercise performed continuously at a submaximal work rate. Both exercise trainings increase individual’s preoperative functional capacity [126,127,128]. Nevertheless, HIIT could better prepare functional adaptation in response to surgical stress: the time spent near the maximal peak capacity (understood as VO_2-peak_) and the maximal increase in stroke volume and musculoskeletal effort may be helpful to withstand surgical stress [129]. In a recent RCT, including 42 prehabilitated patients, Carli et al. compared the effects on perioperative functional capacity (measured by CPET) of two different physical exercise protocols (HIIT n = 21 versus MICT n = 21). Both MICT and HIIT equally improved preoperative functional capacity, but this benefit was maintained only in patients treated with HIIT two months after surgery [126]. Studies conducted in patients undergoing cardiac rehabilitation have shown similar results, confirming that HIIT is a more effective way to improve VO_2-peak_ and cardiac systolic function [130]. Based on these data, HIIT seems a more effective exercise intervention for facilitating functional recovery.

Exercise intensity and protocol (HIIT or MICT) should be based on CPET-derived variables when available, ensuring an individualized and target treatment for each patient [131]. Alternatively, exercise work-rate can be setup using the Borg scale. Furthermore, each session should include both warm-up and cool-down exercises, as well as stretching [132].

#### 3.2.2. Resistance Exercise Training

During resistance training, muscles work against a force. The goal is to increase the muscle strength reserve and the free fat mass index, severely affected by patients’ disease (inflammation and/or catabolism), neoadjuvant cancer therapies, and surgical stress.

During the prehabilitation period, it is essential to gradually increase the volume of exercise for both aerobic and resistance training. This progression can be achieved by increasing either one of the FITT variables; it is recommended to increase frequency and duration before intensity. An example of exercise prescription based on FITT-VP principles during a prehabilitation program is shown in Table 3.

The synergistic effect between physical exercise and protein supplementation facilitates muscle protein synthesis [127,133]. To avoid weight loss during prehabilitation, it is essential that the increased energy expenditure associated with exercise corresponds to an adequate caloric and protein intake.

In many patients, preoperative training must be conducted under professional supervision, especially in frail, sedentary, and non-compliant patients. Supervised exercise training seems more effective than unsupervised exercise for facilitating functional recovery [134]. Supervised training can take place either in hospital-based prehabilitation centres and/or in specialized exercise training facilities located closer to the patient’s home. It should be acknowledged that factors such as transportation, distance to the prehabilitation facility, and time dedicated to training can discourage patients participating in prehabilitation programs [135]. The effectiveness of telemedicine or digital technologies that might permit caregivers to remotely follow the adherence of the prehabilitation program need to be evaluated [136].

There are specific absolute contraindications to exercise training such as severe aortic stenosis, unstable angina, malignant arrythmia, and uncontrolled systolic blood pressure. Addressing these severe co-morbidities is the first priority in the preoperative period in these groups of patients.

#### 3.2.3. Inspiratory Muscle Training (IMT)

IMT, which aims specifically at increasing respiratory muscle strength and endurance, has been shown to reduce postoperative atelectasis and pneumonia and shorten length of hospital stay [137]. This additional prehabilitative intervention might be particularly useful either in CD patients at high-risk of pulmonary complications (CD patients with restrictive respiratory disease ankylosing spondylitis-related) or in CD patients requiring laparotomies (especially upper gastrointestinal surgery), frequent in CD patients requiring multiple surgeries [138]. Typically, patients are asked to breath in through an inspiratory threshold-loading device at a pre-defined percentage of their maximal inspiratory strength (Pimax), 5 to 7 sessions per week, each lasting 15–30 min [137,139].

## 4. Psychological Distress Optimization

### 4.1. Assessment and Monitoring

Three out of four patients with CD undergo surgery [140]. During the preoperative period, CD patients might be worried about hospitalization, separation from family members, disabilities related to the surgery, the process of recovery [141], experiencing psychological distress, as well as anxiety and depression [141,142,143,144,145,146,147,148]. Since psychological distress may impair physical functioning before surgery and poor physical recovery after surgery [143,145,148,149,150,151,152,153], a comprehensive preoperative assessment and personalized treatment might improve outcomes (Figure 2) [148,149,153].

In the preoperative period, psychological distress can be assessed [148,153,154] via clinician- or self-reported measures [155]. Information should be collected also in areas related to psychological distress in medical ill patients, such as illness behaviour (i.e., the ways in which individuals experience, perceive, evaluate, and respond to their health status) [156,157], mental pain (i.e., a feeling state characterized by a sense of hopelessness and/or helplessness, loss of meaning, loss of self, feelings of emptiness, and loss of control or autonomy) [158], beliefs about surgery [159], and psychosomatic syndromes [160,161]. A comprehensive assessment should also consider the interplay between mental and organic diseases in terms of primary/secondary relationship [161,162]. Finally, positive features, i.e., subjective and psychological well-being [163,164], should be considered. This might demarcate major differences among CD patients who otherwise might seem to be deceptively similar since they share the same diagnosis.

### 4.2. Types of Intervention

Interventions aimed at reducing psychological distress in the preoperative period may help CD patients to withstand the stress of surgery, attenuate possible postoperative impairments in psychological and physical function, and accelerate the return to preoperative levels of functioning [149,153,165]. Treatment options include psychoeducation and psychotherapy based on cognitive-behavioural principles [141,148,153,166,167]. Psychoeducation may be used to provide information on the surgical procedure, treatments, expected side effects, recovery process, and functional modifications [141,153]. Cognitive behavioural therapy [168,169] may help patients to focus on the early identification and restructuring of own disruptive beliefs about surgery, which lead to psychological distress [159,168,169], promoting adaptive behaviours [141,153,167]. Intervention aimed at promoting psychological well-being, i.e., Well-Being Therapy [170], may be considered [171]. Well-Being Therapy was found to be promising in helping patients to cope with chronical medical diseases [172,173,174,175].

## 5. Medical Optimization

CD patients might present to surgery with several medical modifiable risk factors that, if timely identified, can be corrected.

Preoperative anaemia is associated with postoperative adverse outcomes [176]. In CD patients, iron-deficiency anaemia, frequently coexisting with anaemia of chronic inflammatory disease, can be corrected or optimized before surgery by IV iron infusion, especially considering the absorption deficit of these patients. The findings of a recent large multicentre randomized controlled trial conducted in surgical patients undergoing major non-cardiac surgery demonstrated that although variations of preoperative haemoglobin concentrations following ferric carboxymaltose infusion might be marginal, postoperatively haemoglobin concentrations continue to raise and remain significantly higher than those of patients not treated with preoperative IV iron. These data suggest that, even when preoperative time is not sufficient to significantly increase haemoglobin concentrations, preoperative anaemia should be still optimized to facilitate surgical recovery, minimize the risk of postoperative complications, and avoid unnecessary allogenic blood transfusions [177].

Similarly, impaired glucose control at the time of major surgery has been associated with postoperative infections, delayed wound healing, and cardiovascular complications [178]. Although the introduction of immunomodulating agent and biologics have significantly reduced the use of corticosteroids, frequently CD patients are still on steroids at the time of surgery, usually tapered down (<15–20 mg of prednisone/day) to prevent infectious complications [179,180]. Several important considerations are required. First, despite gradual reduction of preoperative dosages, glycaemic control of CD patients might be impaired considering the long-lasting chronic exposure to this treatment. Second, to avoid disease flare-up before surgery due to a subtherapeutic steroid dosage, surgery should be planned soon after the end of the prehabilitation period, at the lowest steroid dosage possible. Exclusive enteral nutrition has been used to limit the risk of flare-ups during the preoperative weaning of disease-modifying medications such as corticosteroids. Third, CD patients on a low-dose of steroids should continue their treatment until the morning of surgery. Whether or not administering an additional preoperative stress dose of IV hydrocortisone prevents perioperative adrenal insufficiency, especially when patients are on a low-dose of corticosteroid, remains unclear [181]. Finally, it should be acknowledged that some immunomodulating agent and biologics can affect glucose metabolism [180]. Although, to the best of our knowledge, trials demonstrating that ameliorating preoperative glycaemic control improves postoperative outcomes are lacking, multimodal prehabilitation represents an opportunity to optimize glucose metabolism, and thus minimizing the risk of postoperative complications.

There is ongoing debate regarding the importance of immunosuppressive medications in increasing perioperative risks in CD. A detailed discussion of this topic is beyond the scope of this article. In summary, there is broad consensus that the immunomodulators (azathioprine and 6-mercaptopurine) are unlikely to affect perioperative risks. In contrast, the data on TNF-alpha antagonists such as infliximab are contradictory. There are scant high-quality data on newer classes of drugs such as integrin inhibitors and Janus kinase inhibitors.

## 6. Duration of the Trimodal Prehabilitation Program

The duration of published multimodal prehabilitation programs ranges from 1 to 6 or more weeks [182]. Although prehabilitation studies conducted in colorectal cancer patients typically report an intervention period of 4 weeks, the minimum and optimal duration of a multimodal prehabilitation program remains to be determined. Even if surgery for CD patients is not strictly time-sensitive, and therefore the duration of the prehabilitation program could be prolonged if needed, it should be considered that by excessively postponing surgery to facilitate preoperative optimization might lead to disease flare up and/or to urgent surgery. Therefore, prehabilitation should be individualized and delivered in an optimal time window.

## 7. Conclusions

The physiologic rational supporting multimodal prehabilitation in patients affected by CD before major surgery is provided by several studies demonstrating the efficacy of several heterogenous unimodal interventions (nutritional optimization, medical optimization, inspiratory muscle training) in reducing postoperative complications and facilitating surgical recovery. Preoperative functional capacity and psychological status have been scarcely addressed. It is therefore plausible to hypothesize that the proven benefits of multimodal prehabilitation observed in an heterogenous surgical population undergoing major abdominal surgery, integrating medical, functional, nutritional, and psychological optimization, might be advantageous also for patients with CD. Further studies are needed to establish the efficacy of multimodal prehabilitation in this specific surgical population.

## Figures and Tables

**Figure 1 nutrients-16-01783-f001:**
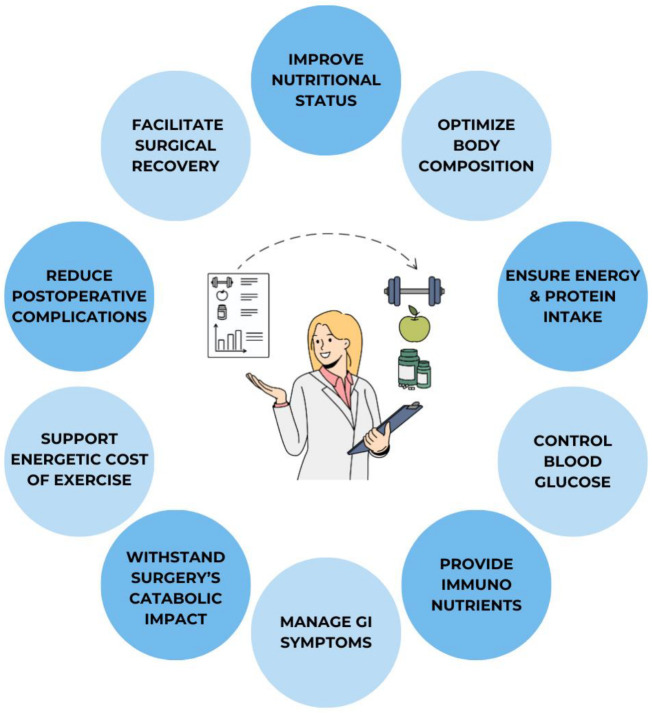
Objectives of perioperative nutritional interventions in patients with inflammatory bowel disease.

**Figure 2 nutrients-16-01783-f002:**
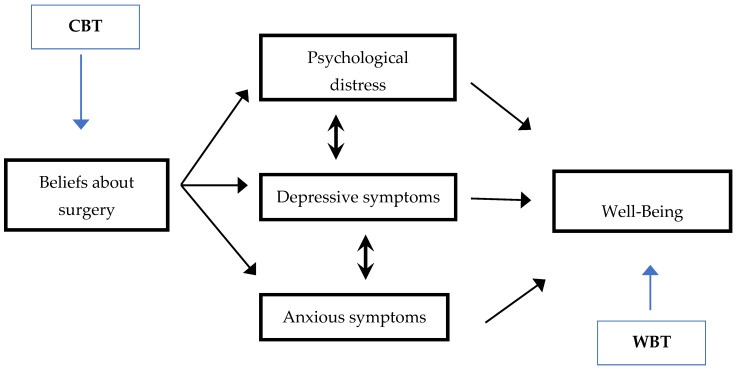
Proposed assessment and therapeutic approaches to preoperative psychological distress: A patient with Crohn’s disease reporting disruptive beliefs about surgery, psychological distress, depressive and anxious symptoms, and poor well-being. The therapist could give priority to cognitive behavioural therapy (CBT) to identify and restructuring disruptive beliefs on surgery in order to reduce psychological distress, depressive and anxious symptoms. Then, the therapist could decide to improve well-being via Well-Being Therapy (WBT) techniques.

**Table 1 nutrients-16-01783-t001:** Preoperative nutritional screening and assessment.

**Preoperative malnutrition Screening**	MUST NRS-2002 MIRT SaskIBD-NR NS-IBD
**Preoperative malnutrition assessment and diagnosis tool**	GLIM Criteria
**Preoperative sarcopenia assessment**	Body composition Muscle Strength (Hand Grip, 5-time chair standing test) Functional assessment (walking tests, gait speed, SPPB, TUG)
**Anaemia: screening and assessment of**	Complete blood count Serum ferritin, transferrin, C-reactive protein, Vitamin B12, Folates, Creatinine
**Micronutrient assessment**	Vitamin B12, Vitamin D, Folic acid

Legend: MUST = Malnutrition Universal Screening Tool; NRS-2002 = Nutrition Risk Screening 2002; MIRT = Malnutrition Inflammation Risk Tool; SaskIBD-NR = Saskatchewan IBD Nutrition Risk Tool; NS-IBD = Inflammatory Bowel Disease-Nutritional Screening tool; GLIM = Global Leadership Initiative on Malnutrition; SPPB = Short Physical Performance Battery; TUG = Timed Up and Go.

**Table 2 nutrients-16-01783-t002:** Predicted Outcomes and Utility of common tools used to measure or estimate preoperative functional capacity in the context of a prehabilitation program.

	Predicted Outcomes and Utility in the Context of a Prehabilitation Program		
	Measurement/Evaluation of Functional Capacity	Predicted Postoperative Outcomes	Utility in the Context of a Prehabilitation Program
**Cardio-Pulmonary Exercise Testing (CPET)**	Gold Standard for measuring cardiopulmonary and musculoskeletal efficiency [83] Measures: VO_2-peak_, VO_2-AT_, VE/VCO_2_, O_2_ pulse [83]	VO_2-Peak_: predicts moderate or severe postoperative complications. [84] VO_2-AT_: high negative predictive values (94–100%) for postoperative mortality [76]	Tailored exercise prescription [85] Evaluating preoperative therapies in cancer surgery [86] Prehabilitation response and risk stratification [87]
**6-Minute Walking Test (6MWT)**	Submaximal exercise test Moderate correlation with VO_2-peak_ [88,89]	Modest association with moderate or severe complications [90]	Prehabilitation response and risk stratification [14] Attaining a 6MWD < 400 m after prehabilitation: higher risk of 30-day postoperative complications [91] Clinically meaningful 6MWD change: ≥20 m [92]
**Duke Activity Status Index (DASI)**	Modest correlation between the DASI score and VO_2-peak_ [93] Moderate ability to predict VO_2-peak_ > 15 mL/kg/min [93]	30-day death, MINS, MI, moderate-to-severe complications, and new disability [94]	
**Subjective assessment of Metabolic Equivalents (METs)**	Low sensitivity for identifying patients with poor functional capacity [81]	Predicts MACE in high-cardiovascular risk population [82] Does not improve MACEs prediction compared with clinical risk factors [95]	-

Legend: MINS = myocardial injury after non-cardiac surgery; MI = myocardial infarction; MACEs = major adverse cardiovascular events; VO_2-peak_ = peak oxygen uptake; VO_2-AT_ = oxygen consumption at the anaerobic threshold; VE/VCO_2_ = Ventilatory equivalent for CO_2_; 6MWD = 6-minute walking distance.

**Table 3 nutrients-16-01783-t003:** Example of Prehabilitation exercise training program based on FITT-VP principles [125].

FITT-VP Exercise Prescription			
	Aerobic Training		Resistance Training
	HIIT	MICT	
**Frequency**	Three times per week (at least 4 weeks)	Three times per week (at least 4 weeks)	Three times per week (at least 4 weeks)
**Intensity**	85–90% VO_2-peak_ Active recovery: 80–85% VO_2_AT	80–85% VO_2 AT_	60–80% of 1RM
**Time**	34 min (including 5 min of warm-up and 5 min of recovery)	40 min	30 min
**Type**	Cycle ergometer, treadmill, NuStep	Cycle ergometer, treadmill, NuStep	Dumbbell, elastic band, stick, med ball
**Volume**	4 repetitions of high intensity (3 min) with active rest (4 min)	Continuous	3 progressive sets (e.g., 10 × 3, 12 × 3, 15 × 3) of upper, lower, total body and abdominals
**Progression**	Monthly cycles	Monthly cycles	Weekly cycles

Legend: VO_2_ = Oxygen Uptake; AT = Anaerobic Threshold; HIIT = High Intensity Interval Training; MICT = Moderate Intensity Continuous Training; 1RM = one-repetition maximum test.

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
