# Peer review of "Multimodal Prehabilitation for Patients with Crohn’s Disease Scheduled for Major Surgery: A Narrative Review"

_nutrients, 2024, doi:10.3390/nu16111783_

Round 1
Reviewer 1 Report
Comments and Suggestions for Authors
The authors have chosen to discuss in this review a subject as difficult as it is challenging to address in clinical practice. Surgery in patients with Crohn's disease represents a very important tool in their management, which puts additional pressure on the patient. The authors have managed to provide important information regarding the generalities related to Crohn's disease surgery, as well as important data regarding what prehabilitation entails, an umbrella term that encompasses a multitude of intervention modalities to ensure the best possible post-surgical prognosis. Although it is a difficult subject, the authors have succeeded in gathering a multitude of references to create a high-quality review.
Minor comments:
The authors could include in the introduction section some data about physical inactivity as a debilitating issue in patients with IBD, particularly those with Crohn's disease.
Comments on the Quality of English LanguageMinor review
Reviewer 2 Report
Comments and Suggestions for Authors
The Italian authors attempted in this review article to present the physiologic rationale supporting multimodal “prehabilitation”, that is to improve the preoperative medical, physical, nutritional and psychological status, in patients with Crohn’s disease (CD) waiting for surgery. My concerns are as follows.
1. There remained some careless mistakes, including typos. I was not able to find any figure below “Figure 1. Objectives of perioperative nutritional interventions in patients with inflammatory bowel 204 disease”. I truly hope the authors to revise this manuscript very carefully.
2. The original terms of all abbreviations appearing in two tables are better to be added below the tables. “VO2: Oxygen Uptake; AT: Anaerobic Threshold; HIIT: High Intensity Interval Training; MICT: Moderate Intensity Continuous Training; 1RM: one-repetition maximum test” are better to be moved to the bottom of Table 2.
3. The data shown in Table 2 need to be added the numbers of references based on.
4. The legend(Note) of Figure 2 is better to be separate from the main text.
5. I wonder whether the multimodal prehabilitation having been observed in the heterogenous surgical populations undergoing major abdominal surgeries, integrating medical, functional, nutritional and psychological optimization, have been proven to be advantageous for CD patients undergoing surgery based on enough literatures.
Comments on the Quality of English LanguageNil
Reviewer 3 Report
Comments and Suggestions for Authors
Thank you very much for the opportunity to review the article.
The topic of reporting is extremely important, but it has been presented many times in the literature in a much more systematic way.
The article is a review of the literature, but there is no information on how the article was selected for the review.
There are articles that are over 10 or even 20 years old.
The abstract uses abbreviations and the text contains numerous linguistic errors.
Thank you very much for the opportunity to review the article.
The topic of reporting is extremely important, but it has been presented many times in the literature in a much more systematic way.
The article is a review of the literature, but there is no information on how the article was selected for the review.
There are articles that are over 10 or even 20 years old.
The abstract uses abbreviations and the text contains numerous linguistic errors.
I do not recommend the article for publication.
Round 2
Reviewer 2 Report
Comments and Suggestions for Authors
I still doubt that the authors had successfully shown enough data or references to support their consideration.
Comments on the Quality of English LanguageNil
Reviewer 3 Report
Comments and Suggestions for Authors
The authors did not respond satisfactorily to the comments presented